# Enhancing Silicon Solar Cell Performance Using a Thin-Film-like Aluminum Nanoparticle Surface Layer

**DOI:** 10.3390/nano14040324

**Published:** 2024-02-06

**Authors:** Mirjam D. Fjell, John Benjamin Lothe, Naomi J. Halas, Mali H. Rosnes, Bodil Holst, Martin M. Greve

**Affiliations:** 1Department of Physics and Technology, University of Bergen, P.O. Box 7803, 5020 Bergen, Norway; mirjam.dyrhovden@uib.no (M.D.F.); john.lothe@uib.no (J.B.L.); bodil.holst@uib.no (B.H.); 2Department of Electrical and Computer Engineering, Rice University, Houston, TX 77005, USA; halas@rice.edu; 3Department of Chemistry, University of Bergen, P.O. Box 7803, 5020 Bergen, Norway; mali.rosnes@uib.no

**Keywords:** renewable energy, light in-coupling, plasmonics

## Abstract

Solar cells play an increasing role in global electricity production, and it is critical to maximize their conversion efficiency to ensure the highest possible production. The number of photons entering the absorbing layer of the solar cell plays an important role in achieving a high conversion efficiency. Metal nanoparticles supporting localized surface plasmon resonances (LSPRs) have for years been suggested for increasing light in-coupling for solar cell applications. However, most studies have focused on materials exhibiting strong LSPRs, which often come with the drawback of considerable light absorption within the solar spectrum, limiting their applications and widespread use. Recently, aluminum (Al) nanoparticles have gained increasing interest due to their tuneable LSPRs in the ultraviolet and visible regions of the spectrum. In this study, we present an ideal configuration for maximizing light in-coupling into a standard textured crystalline silicon (c-Si) solar cell by determining the optimal Al nanoparticle and anti-reflection coating (ARC) parameters. The best-case parameters increase the number of photons absorbed by up to 3.3%. We give a complete description of the dominating light–matter interaction mechanisms leading to the enhancement and reveal that the increase is due to the nanoparticles optically exhibiting both particle- and thin-film characteristics, which has not been demonstrated in earlier works.

## 1. Introduction

Crystalline silicon (c-Si) solar cells dominate the solar cell market today, accounting for more than 90% of the market [1]. The efficiency of solar cells is directly correlated to the number of photons transmitted into and absorbed by the absorbing layer of the cells [2]. Since c-Si is a highly reflective material, methods for reducing light reflection are critical. The two main techniques employed in commercial c-Si solar cells are anti-reflection coatings (ARCs) and surface texturing. ARCs are typically transparent dielectric materials. The thickness and refractive index of the ARC are designed and selected to give destructive interference within a certain wavelength range for the reflected light to maximize the number of photons transmitted [3]. By texturing the surface of the cell, and making it corrugated, two additional benefits are obtained with respect to light coupling into the cell. First, light reflecting off the textured surface will have a higher probability of being transmitted into the active layer due to multiple reflections. Second, if the surface is textured by some random geometry, it becomes more like a Lambertian surface, which scatters light with equal intensity in all directions. The scattered light will increase the path length of the transmitted light, hence also increasing the probability of charge carrier generation [4,5]. For a specific solar cell slab, with its inherent quantum efficiency, the number of photons absorbed by the cell is determined by the optical properties of the cell itself and its material-specific optical properties. Using these types of light trapping schemes, it has been shown that the upper light absorption enhancement is limited by $4n^{2}$ relative to single-pass light absorption, where *n* is the refractive index of Si [6]. This limit is sometimes referred to as the traditional ray optics limit and is found using simple ray tracing methods and length scales much longer than the wavelength of light.

However, by describing the light–matter interaction using electromagnetic models, the discussion can be extended to sub-wavelength and nanophotonic systems. In doing so, light-trapping mechanisms that go beyond the traditional ray optics limit, using sub-wavelength structures such as nanoparticles and photonic crystals, have been proposed [7,8]. Several studies using metallic nanoparticles to further increase the performance of solar cells have been reported over the years [9,10,11,12,13,14,15,16]. Metallic nanoparticles scatter and absorb light efficiently within specific spectral regions due to their localized surface plasmon resonances (LSPRs). The LSPR also leads to a near-field concentration of light, and these effects can be exploited to enhance light absorption in solar cells [17,18]. The LSPR wavelength is strongly dependent on the nanoparticle material, size, and shape as well as on its surrounding medium, which are all parameters that can be easily modified.

The use of gold (Au) and silver (Ag) nanoparticles on solar cells has been widely studied, and promising enhancements have been demonstrated [19,20,21,22,23,24]. The LSPR of both Au and Ag nanoparticles can be tuned to the visible spectral region, thus increasing scattering into the solar cell in this region. However, there are several concerns with these materials: parasitic light absorption in the particles for wavelengths around the LSPR wavelength reduces the amount of light scattered into the solar cells [25]. Additionally, the coinage metal Au is rare and expensive, which severely limits its use in any mainstream commercial applications.

Thus, to successfully employ nanoparticles for enhancing light coupling into solar cells, we identify three initial criteria that must be met: (1) the material must exhibit plasmonic activity within the spectral range of interest with forward scattering as the dominating light-scattering direction; (2) the material should have minimal light absorption in the wavelength range of interest; and (3) the material should be abundant and low-cost.

One material fulfilling these criteria is aluminum (Al). Al is an earth-abundant and inexpensive material, and Al nanoparticles have a tuneable LSPR in the UV to visible spectral region [26]. Thus, parasitic light absorption by the nanoparticles in the solar spectrum can be minimized [25]. Al nanoparticles have been investigated for use in both c-Si solar cells [27,28,29] and thin-film solar cells [30,31,32,33]. However, none of these studies give a complete description of the underlying mechanisms yielding increased solar cell efficiency, and it is therefore not clear how the light in-coupling is obtained. However, this information is necessary to optimize the system for maximum performance. Parashar et al. [34] studied experimentally how the native Al oxide (Al_2_O_3_) layer that forms on Al nanoparticles affects the performance of a flat c-Si solar cell. A simplified system with c-Si solar cells without an ARC was used and the efficiency increased from 8.80% without nanoparticles to 9.17% with nanoparticles without an oxide layer, and further to 10.42% when the nanoparticles had a 2.17 nm thick oxide layer. Zhang et al. [35] found that spherical Al nanoparticles placed on top of a flat c-Si solar cell increase the transmittance of light into the cell over a broad wavelength range in the solar spectrum. In their work, simulations showed that for a flat c-Si solar cell with an 80 nm thick SiN_x_ ARC, referred to as a spacing layer in the paper, the light transmittance increased by approximately 2% when the Al nanoparticles were placed on top of the ARC.

To the best of our knowledge, there has been no systematic study on the tuning and optimization of the key parameters of Al nanoparticle performance for applications on a commercial standard textured c-Si solar cell. Further, the predominant light in-coupling mechanisms have also not been fully understood, which clearly are of key importance when trying to maximize the amount of light entering the cell. In this work, we present a thorough study of spherical Al nanoparticles on flat c-Si solar cells with a commercial standard ARC to optimize the nanoparticle parameters through numerical simulations and give the full description of the underlying mechanisms for the light–matter interactions. This has not been done before as far as we are aware. We have varied and optimized all nanoparticle parameters to uncover the highest light transmission into the solar cell. For the best particle parameters, we have tuned the thickness of the ARC layer for an optimal cell design. Finally, we have also, with promising results, simulated the transmission for light incident at an angle, indicating that our new optimized parameters will also work for textured commercial standard c-Si solar cells.

## 2. Materials and Methods

The Finite Difference Time Domain (FDTD) method, which solves Maxwell’s equations for complex geometries, has been used to simulate the transmission of light into a flat semi-infinite c-Si wafer coated with a silicon nitride (Si_3_N_4_) layer as the ARC. The FDTD simulation technique is not suited to simulating the micron-scaled textured surface of a commercial standard c-Si solar cell together with the nano-scale nanoparticles on the surface as it would require a large simulation region with high mesh accuracy to resolve the nanoparticles. This is both time- and capacity-consuming, and we therefore simplified the system to a flat surface with nanoparticles on top for the parameter optimization process. The Ansys Lumerical FDTD simulation software, version 2021 R1.2, was used to perform the simulations. The simulation setup is illustrated in Figure 1. The thickness of the ARC was initially set to 75 nm in our simulations, which is the optimum single-layer Si_3_N_4_ ARC thickness and is commonly used for c-Si solar cells [36]. Al nanoparticles were placed in a periodic array on top of the Si_3_N_4_ layer in order to simulate a large sample. To obtain a periodic array, periodic boundary conditions (BCs) were used in the x and y directions. In the direction of wave propagation, perfectly matched layer (PML) BCs were used to avoid reflection from the boundaries. As Al oxidizes under ambient conditions, a 3 nm Al_2_O_3_ oxide layer was used in the initial simulations, as we expect this to always be present in an experimental setting [37]. The material data for Al, Al_2_O_3_, and c-Si were found in Palik [38], while the material data for Si_3_N_4_ were found in Philipp [39]. The mesh grid used was a conformal variant, which adjusts the mesh to the refractive index of the materials. In addition, a mesh override region with a grid size of 1 nm in the x, y, and z directions was used on the nanoparticles to ensure the nanoparticles and oxide layers were well resolved. The mesh override region extended three grid cells outside the nanoparticles. The mesh size was determined through convergence testing. A plane wave light source was used with a wavelength interval of 300–1130 nm, where the shortest wavelength matches the solar spectrum and the longest corresponds to the band gap of Si [40]. The light source was placed at least ½$\lambda_{max}$ above the nanoparticles. The angle of incidence of the light, θ, was set to zero. The monitor measuring the transmission of light into c-Si was placed 1 nm into the c-Si layer. The transmission results were weighted to the AM 1.5 solar spectrum [41]. Since the number of electron–hole pairs generated in the solar cell is directly correlated with the number of photons transmitted into the c-Si, the number of photons transmitted was calculated by integration [2].

Based on this simulation setup, an iterative optimization process for maximizing light in-coupling into the c-Si was carried out as follows. First, the ideal nanoparticle size was determined. Next, the effect of nanoparticle surface coverage and oxide thickness of the nanoparticle oxide layer was investigated. Finally, the ARC thickness was tuned in order to maximize the device performance for the added nanoparticle layer.

To demonstrate that our results are also reliable for a textured commercial standard c-Si solar cell, we tilted the light source relative to the surface normal and simulated the transmission of light into c-Si for three angles θ: 10°, 30°, and 54.7°. The 54.7° angle imitates light hitting a textured solar cell at normal incidence, as the angle is the same as the base angle of the pyramids that arise when the c-Si is etched during solar cell production [42]. It should be noted that this approach does not take into account multiple reflections between the pyramids and therefore does not fully represent a textured solar cell. Still, it gives us valuable insight into the transmission of light at each pyramid depending on the incident angle. The Broadband Fixed Angle Source Technique (BFAST) provided by Ansys Lumerical FDTD was used to simulate the light transmission when the light was incident at a non-normal angle. In this technique, the FDTD update equations are reformulated using the field transformation method, thereby removing the wavelength dependence of the incident angle [43]. Due to the reformulation in the FDTD algorithm, the simulation time increases significantly when using BFAST. This technique uses built-in periodic BCs in the x and y directions, while PML BCs were used in the z direction. The transmission for both p- and s-polarized light was simulated, and then the transmission for unpolarized light was calculated. The simulations were carried out for the bare solar cell and the solar cell with optimized parameters.

In addition, to determine the scattering and absorption cross sections of the nanoparticles, simulations using a total-field-scattered-field (TFSF) light source box were carried out. The TFSF light source box surrounded a nanoparticle in one unit cell, with its x and y boundaries extending outside the simulation region. Again, periodic BCs were used in x and y and PML in the direction of the wave propagation. The absorption cross section was measured with monitors surrounding the nanoparticle inside the light source box, while the scattering cross section was measured with monitors above and below the light source box. To determine the angular scattering distribution, simulations of a single nanoparticle with a simulation region much greater (50×) than the nanoparticle dimensions were carried out. A TFSF light source box surrounded the nanoparticle, and PML boundaries were used in all directions. Monitors extending outside the simulation region were placed above and below the light source box.

## 3. Results

### 3.1. Size of Al Nanoparticles

Al nanoparticles with diameters ranging from 60–180 nm were studied. Nanoparticles smaller than 60 nm were not studied since their LSPR wavelength lies in the far UV, hence the effect they will have on the solar cell is small [44]. An initial surface coverage of 8.7% was used, corresponding to three times the diameter of the nanoparticles. This surface coverage was chosen initially to eliminate near-field effects between the nanoparticles. Figure 2 shows the normalized number of photons transmitted into c-Si as a function of the nanoparticle size. The normalized number of photons, n, is given by:(1)$$n=\frac{T_{\mathrm{NP}}}{T_{\mathrm{bare}}},$$
where $T_{\mathrm{NP}}$ is the number of photons transmitted into the solar cell with nanoparticles present and $T_{\mathrm{bare}}$ is the number of photons transmitted into a bare solar cell, that is, a solar cell with the standard 75 nm ARC without nanoparticles present. We found that all nanoparticle sizes except the largest (180 nm) give an enhancement in the transmission of photons into c-Si. Since, however, the increase in transmission for the 60 nm and 160 nm nanoparticles is small, only nanoparticles with sizes ranging from 80–140 nm were studied further.

### 3.2. Surface Coverage of Al Nanoparticles

We studied the Al nanoparticles for various inter-particle spacings to determine the optimized nanoparticle surface coverage. Intuitively, based on the results from Section 3.1, it is expected that increasing the surface coverage will result in more light in-coupling. However, denser nanoparticle arrays will also exhibit increased light absorption and possibly near-field effects. In addition, there is the effect of the periodicity of the array itself which will give rise to periodic interference phenomena in the far field.

The inter-particle spacings studied ranged from 1.5*d*–6.5*d*, where *d* is the diameter of the respective nanoparticles. This corresponds to a surface coverage ranging from 34.9–1.9%. The corresponding interparticle distances in nm are shown in Table 1. In Figure 3, the normalized number of photons transmitted into c-Si is shown as a function of the surface coverage. A surface coverage below 21.0% gives an increase in the transmission for all nanoparticle sizes. Further, it is interesting to note that the same surface coverage of 12.6% results in the highest transmission increase independent of the nanoparticle sizes, and therefore also the inter-particle spacing. The nanoparticles giving the largest enhancement in the transmission of light are the nanoparticles with a diameter of 120 nm, yielding an enhancement of 2.39%.

### 3.3. Oxide Layer Thickness of Al Nanoparticles

As Al oxidizes quickly in ambient air it is important to investigate how the oxidation of Al nanoparticles affects the transmission. Hence, simulations with different oxide thicknesses were performed. The oxide thickness varied from zero (pristine Al nanoparticles) to fully oxidized nanoparticles (Al_2_O_3_ nanoparticles), in steps of 10% of the radius of the respective nanoparticle size. At this point, the surface coverage was kept at 12.6%. The results are presented in Figure 4. For the three smallest nanoparticle sizes (80, 100, and 120 nm), an enhancement in the transmission is observed when the oxide layer is introduced and increased, compared to without the oxide layer. However, when the thickness of the oxide layer continues to increase, the transmission eventually reduces even below that of the pristine Al nanoparticles. However, the transmission for the largest nanoparticles (140 nm) is quite different. Here, the transmission decreases for thin oxide layers before it increases and surpasses the transmission for all the other nanoparticle sizes, with a peak at an oxide thickness of 42 nm (corresponding to 60% of the radius) where the enhancement is 2.61%. This clearly shows that the transmission of light is highly dependent on the oxide thickness of the nanoparticles and that the effect of the oxide thickness is dependent on the nanoparticle size. Based on these results, new simulations with varying surface coverage were performed for the nanoparticles with a diameter of 140 nm and an oxide thickness of 42 nm. These showed that the ideal surface coverage was now 21.0%, yielding a 3.14% enhancement in the transmission of light into c-Si (see Appendix A).

### 3.4. ARC Thickness

After optimizing nanoparticle size, surface coverage, and oxide thickness, the thickness of the ARC was varied with the ideal particle configuration on top to find the optimal ARC thickness when nanoparticles are present. The thickness was varied from 60–100 nm, with increments of 5 nm. The results are presented in Figure 5. Note that when calculating the normalized number of photons transmitted (Equation (Equation 1)), $T_{\mathrm{bare}}$ is the number of photons transmitted into c-Si with a 75 nm thick ARC as this is the commercial standard. The ARC thickness strongly affects the transmission of light into c-Si. This is as expected since the thickness of the ARC determines the wavelengths at which it is most effective. From Figure 5 it can be seen that when the nanoparticles are present all ARC thicknesses studied herein result in an enhancement in the light transmission. However, the optimal thickness for the ARC shifts from 75 nm to 80 nm, and the total increase in transmission is 3.33%. Considering a solar cell with optimized material properties, the total increase in transmission of light into c-Si corresponds to an absolute increase in solar cell efficiency of 1.0% (see calculation in the Appendix A). For the remainder of the paper, only the optimized nanoparticle and ARC parameters will be considered.

### 3.5. Angled Illumination

After completing the optimization process, simulations with a non-normal light incident angle were performed as described in Section 2. Figure 6 shows the normalized number of photons transmitted into c-Si as a function of incident light angle. Zero degrees corresponds to the flat substrate and gives the highest transmission. The transmission reduces as the light incident angle increases until the angle of 54.7°, which in turn corresponds to normal incidence on a textured c-Si solar cell surface. As is clear from the results, even though the transmission reduces as the incident light angle increases, all angles yield an enhancement in light transmission and never drop below 2.6%.

## 4. Discussion

### 4.1. Light-Matter Interaction

To understand the underlying mechanisms that yield the enhancement in transmission, we take a closer look at how the Al nanoparticle system interacts with light. The system can be seen from three different perspectives: (1) The interactions of the nanoparticles themselves; (2) the effects due to the periodic arrangement of the nanoparticles; and (3) the optically changed layer above the ARC due to the nanoparticles.

From the first perspective, we look at the scattering and absorption cross section plots for the optimized system, shown in Figure 7. The cross section is the ratio of the total absorbed/scattered power to the power per unit area of the incident light, and is normalized by dividing by the geometric area of the nanoparticle. The scattering cross section plot (dark blue) peaks at a wavelength of 386 nm. This wavelength corresponds to the dipole LSPR wavelength of the nanoparticle [44]. The splitting of the peak is a periodic effect and is not seen for a single particle. As can be seen from this plot, the LSPR comprises an envelope of forward (light blue, solid) and backward (light blue, dashed) scattering components of similar magnitudes, but shifted in peak position wavelength. The nature of these two contributions can be explained by the phase relation of the LSPR and the incident and reflected light fields. Across the LSPR frequency, the LSPR phase will change its sign from ±ϕ to ∓ϕ and, consequently, any destructive/constructive interference between the light source and the reflected light will become constructive/destructive, respectively [17]. This phenomenon is commonly known as Fano resonance [45]. For wavelengths below 369 nm the backward scattering dominates, a domain where the phase of the LSPR is constructive with the reflected light field. Above 369 nm, forward scattering mostly dominates, except in the interval 400–415 nm, where the backward and forward scattering are equal. The absorption cross section plot (purple) has a noticeable component for short wavelengths, peaking at a wavelength of 340 nm. However, in this wavelength range, the number of photons in the solar spectrum is low, hence light absorption by the particles is less critical. As the wavelength increases, the absorption becomes negligible. Both the scattering and the absorption cross section plots have a tail extending far into the visible spectral range, and the scattering takes on a value larger than the absorption for all wavelengths.

Addressing the second perspective, the nanoparticles will act as secondary sources of scattered light after their interaction with the incoming light and, in a periodic arrangement, diffraction and interference effects are expected to occur in the far field. For a broadband light source, however, a distinct diffraction pattern will not be seen in the transmission, but rather interference signals from several interferences such as diffraction, in-plane Wood-Rayleigh anomalies, and the already mentioned Fano resonances [46,47].

Finally, and perhaps less intuitively in this context, the nanoparticle array may also exhibit interference effects similar to those of a thin-film, as the incident light field will experience an overall refractive index change. Since the overall refractive index change will be directly correlated with particle density, this effect will become increasingly prominent with increasing surface coverage. In this case, the interference effect of a nanoparticle array can be described by an effective permittivity which can be calculated using effective medium theory (EMT). Using the Maxwell Garnett approximation, the effective permittivity $\varepsilon_{eff}$ is given by:(2)$$\varepsilon_{eff}=\varepsilon_{s}\left(1+3f_{v}\frac{\varepsilon_{i}-\varepsilon_{s}}{\varepsilon_{i}+2\varepsilon_{s}-f_{v}\left(\varepsilon_{s}-\varepsilon_{i}\right)}\right),$$
where $\varepsilon_{s}$ is the permittivity of air, $\varepsilon_{i}$ is the permittivity of the Al nanoparticles, and $f_{v}$ is the volume fraction of the Al nanoparticles in the air–nanoparticle composite [48]. The permittivity of the Al nanoparticles can be calculated using Bruggeman’s model [49],
(3)$$f_{v,1}\frac{\varepsilon_{1}-\varepsilon_{i}}{\varepsilon_{1}+2\varepsilon_{i}}+f_{v,2}\frac{\varepsilon_{2}-\varepsilon_{i}}{\varepsilon_{2}+2\varepsilon_{i}}=0,$$
where $f_{v,1}$ and $f_{v,2}$ are the volume fractions of Al and Al_2_O_3_ inside the nanoparticle, and $\varepsilon_{1}$ and $\varepsilon_{2}$ are the permittivities of Al and Al_2_O_3_, respectively. Using Equations (Equation 2) and (Equation 3), the solar cell with nanoparticles can be simplified as a stack of three films: the nanoparticle layer, the ARC, the c-Si, and the transmission spectrum can be calculated as T = 1 − R − A using the Fresnel equations [50], where T is the transmission, R is the reflection, and A is the absorption.

### 4.2. Spectral Response for Optimized Device Design

The light source independent transmitted spectrum for the optimized parameters is presented and compared with the flat solar cell with the standard 75 nm ARC in Figure 8a. We see that the added Al nanoparticles and changed ARC thickness give an increase in transmission for most wavelengths, except in two loss regions: below 360 nm, and in the interval 510–680 nm. To better visualize the effect of these increasing and decreasing regions under solar conditions, the difference in the number of photons transmitted with and without optimized nanoparticles, using the solar spectrum, is presented in Figure 8b. As can be seen, the loss below 360 nm is relatively small as there are fewer photons in this spectral range of the solar spectrum. For the second loss range, 510–680 nm, the decrease in transmission yield is more significant as this is in the central part of the solar spectrum with a higher photon flux.

The nanoparticle transmission spectrum in Figure 8a has several important regions. In the shorter wavelength range the effects of the LSPRs of the particles are most important. Around 340 nm, a prominent dip in transmission can be seen. This dip is caused by the LSPR-related light absorption and back-scattering of light (see purple curve in Figure 7). The small peak seen at about 380 nm corresponds to the increased forward scattering of light due to the dipolar LSPR mode of the nanoparticles (see solid light blue curve in Figure 7). It appears small, but when compared to a bare c-Si substrate with 80 nm ARC, exactly at the LSPR peak, the transmission is in fact increased by ≈43% from 0.49 to 0.70.

To the left of the 340 nm dip in Figure 8a, a steep increase in the transmission spectra is seen. This increase does in fact continue outside the spectra and is mainly caused by thin-film interference effects due to the overall refractive index change caused by the nanoparticle layer (see Appendix A). Additionally, the quadrupolar mode of the nanoparticle LSPR located in the UV range further contributes to this increase [51].

On the right side of the nanoparticle LSPR (≈380 nm), the reduction in transmission in the wavelength interval of 510–680 nm is attributed to the change in the ARC thickness and also to the reduced efficiency of the ARC due to nanoparticle scattering. As the ARC thickness increases, the wavelengths at which it is most efficient red shift. Consequently, the transmission at the wavelengths where the standard ARC is most efficient, around 600 nm, will be reduced when the ARC thickness is changed. In addition, for the ARC to function optimally, the light must hit the substrate interface at normal or close to normal incidence [52]. Figure 8c shows the angular scattering by a single nanoparticle at a wavelength of 640 nm where an 80 nm thick ARC is designed to be most efficient. It is clear that parts of the light that are scattered by the nanoparticles will not hit the substrate interface at normal incidence and, consequently, we expect that the efficiency of the ARC is reduced.

Finally, above 680 nm, the nanoparticles barely scatter light, and very little of the transmission spectrum can be attributed to the LSPR of the particles (see Figure 7). However, the particles obviously constitute a layer with a refractive index different from air. Figure 8d compares a numerically calculated transmission spectrum using Fresnel equations with EMT (Equations (Equation 2) and (Equation 3)) for a bi-layer thin-film on a solar cell (blue curve) with the FDTD simulation for the nanoparticles with optimized parameters (orange curve). Even though there is not a perfect match, a close agreement can be seen between the two curves above 500 nm. Hence, the increase in transmission for the longer wavelengths is due to thin-film interference effects due to the nanoparticle layer. A significant deviation is seen between the simulated transmission spectrum and the EMT model from 500 nm and below. This discrepancy is caused by the Al nanoparticle LSPR-induced forward and backward scattering and absorption (see Figure 7), which are not accounted for in the EMT model.

To summarize, the transmission of light into c-Si results from a combination of the light scattered and absorbed by the nanoparticles and the thin-film interference effect caused by the nanoparticle layer. Interestingly, the largest contribution to the increased transmission comes from the thin-film interference effect and not from the particle interactions themselves, as this contribution lies in the spectral region with most photons. This means that the periodicity of the nanoparticle array contributes less to the increased transmission, and we expect that a random array would give equal, if not better, transmission results for the same surface coverage. This is important in an experimental realization.

To minimize reflectance losses, standard c-Si solar cells are textured and not flat as simulated in this work. The combination of such micro and nano-structured geometries is difficult to simulate and would be extremely time-consuming. To demonstrate that our optimization also works for textured cells, we approach the problem by simulating our optimized configuration using angled incident illumination (see Section 2). We find that, for light incident at an angle, we also get a substantially increased light in-coupling. However, we cannot fully account for all light interactions, e.g., multiple reflections, in such a simulation setup. The multiple reflections are particularly important for the light management in solar cells and must be considered when modeling the complete system. In this discussion, we will treat the pyramidal structures to work “as normal”, and limit the discussion to the changes we introduce.

When adding the nanoparticles, the light–nanoparticle interaction is either light scattering or absorption. From Figure 7, we can see that the plasmonic activity of the nanoparticles is in the UV–near-UV region of the spectrum, and the plasmon-related losses are therefore expected to be limited to this spectral region. At the plasmon peak, the distribution of forward and backward scattering is about 50%, and the light absorption is at a maximum for slightly shorter wavelengths. For any wavelengths longer than the resonance wavelength, this ratio is favorable for the forward scattering direction, and back-scattering and absorption losses will quickly diminish (except in the small interval of 400–415 nm). To estimate the transmission losses versus the gains when adding nanoparticles, the absorbed light represents a pure loss channel. For the back-scattered light, it can be expected that some portion of the scattered light will still hit a neighboring pyramid and can be transmitted there. However, due to the angled illumination, the distribution of the back-scattered light will no longer be symmetrical as shown in Figure 8c. Instead, slightly more of the light will be scattered in the direction opposite the incident light (see Appendix A), away from the solar cell’s surface. Hence, some of the back-scattered light will be lost. However, the overall losses due to particle absorption and scattering are limited. Around the plasmon resonance, the solar spectrum has very few photons, making losses at these wavelengths less significant. For longer wavelengths, where there are more photons, the nanoparticle back-scattering and absorption diminish and become negligible. The pyramids then work as normal, but with a double layer ARC comprising the Si_3_N_4_ layer and the nanoparticle layer, which further reduces reflection. Based on these results and assumptions it is therefore fair to assume that the optimized parameters should also provide a positive contribution to a textured solar cell.

## 5. Conclusions

In this work, we have investigated the physical parameters of Al nanoparticles and an ARC in order to maximize light in-coupling in standard c-Si solar cells. We achieve an enhancement in light transmission of up to 3.3% with our optimized parameters: a nanoparticle diameter of 140 nm, an oxide layer thickness of 42 nm, a surface coverage of 21.0%, and an ARC thickness of 80 nm. We demonstrate that the optical transmission gains arise from the nanoparticle array exhibiting both particle characteristics and interference effects similar to that of a thin-film. We attempt to address the standard textured cell surface and argue that our proposed nanoparticle array will also work well for such cells. Our work reveals a novel, experimentally realizable, and efficient system for increasing light in-coupling in solar cells by adding a dispersed layer of tailored nanoparticles on their surface. The work opens the path for several other applications and material systems.

## Figures and Tables

**Figure 1 nanomaterials-14-00324-f001:**
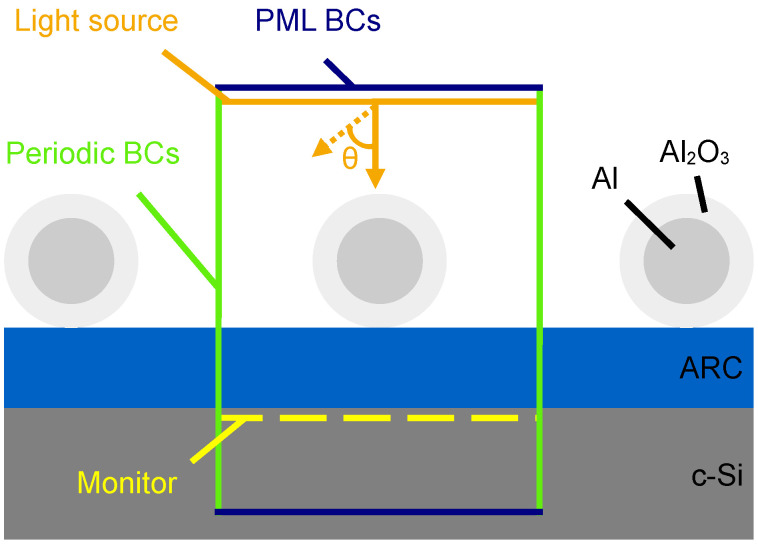
Schematic representation of the simulation setup of the solar cell structure investigated. Not to scale.

**Figure 2 nanomaterials-14-00324-f002:**
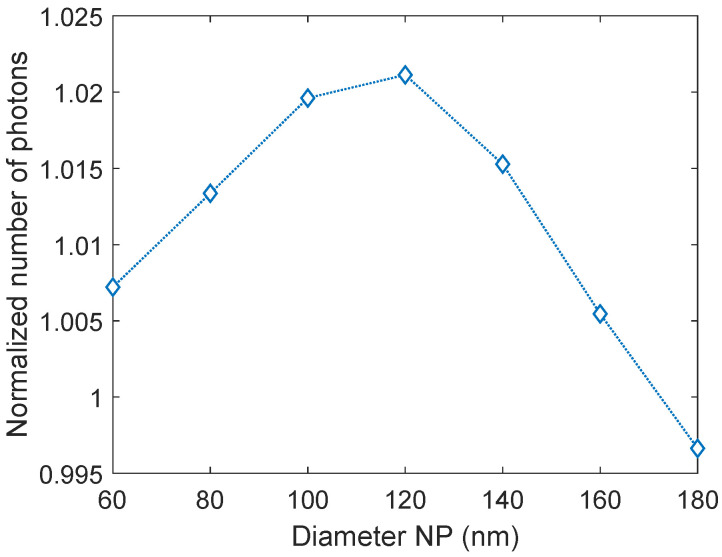
The normalized number of photons transmitted into crystalline silicon (c-Si) as a function of aluminum (Al) nanoparticle diameter. All nanoparticle sizes except 180 nm yield an increase in light transmission.

**Figure 3 nanomaterials-14-00324-f003:**
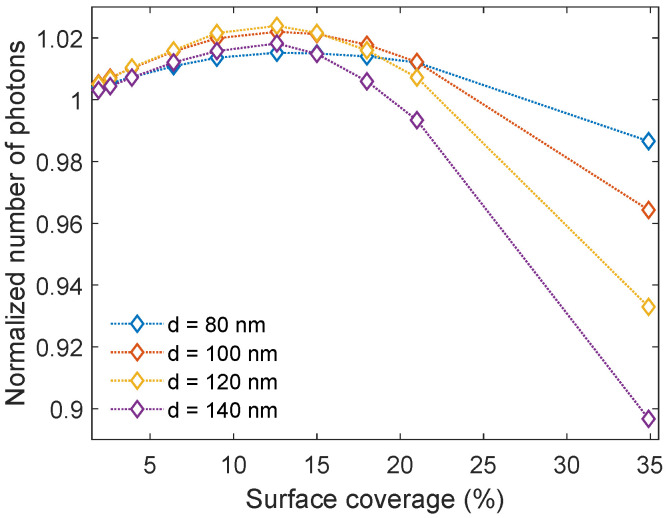
The normalized number of photons transmitted into c-Si as a function of surface coverage. All surface coverages below 21.0% give an enhancement in light transmission, and all nanoparticle sizes exhibit the largest enhancement for a surface coverage of 12.6%.

**Figure 4 nanomaterials-14-00324-f004:**
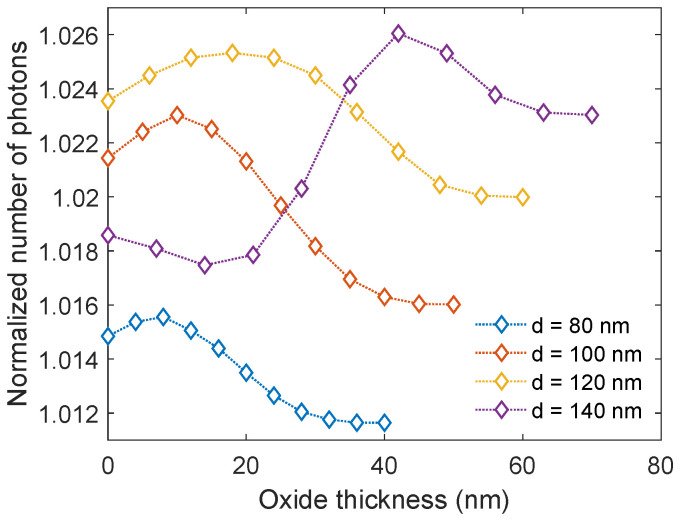
The normalized number of photons transmitted into c-Si as a function of wavelength for various oxide thicknesses. The surface coverage is 12.6%. The nanoparticles with a diameter of 140 nm give the highest transmission.

**Figure 5 nanomaterials-14-00324-f005:**
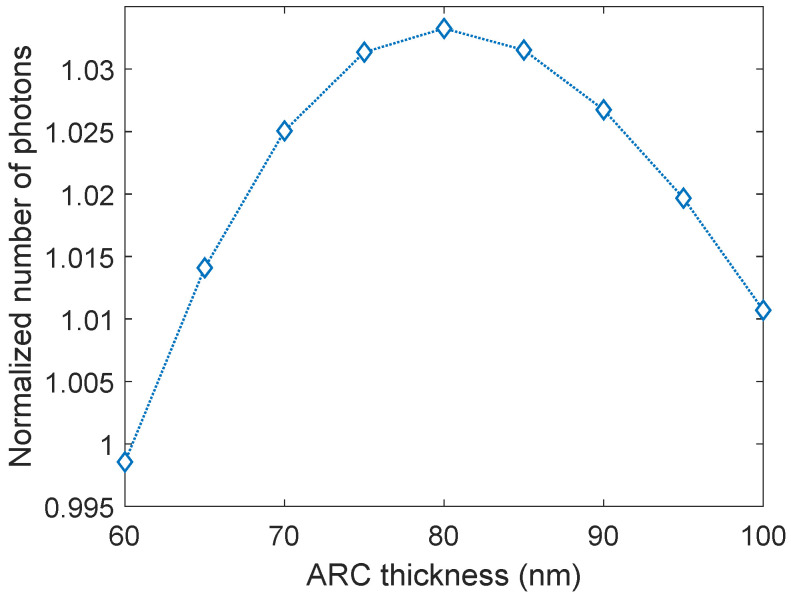
The normalized number of photons transmitted into c-Si as a function of anti-reflection coating (ARC) thickness. The nanoparticles on top of the ARC have a diameter of 140 nm, a 42 nm oxide thickness, and a surface coverage of 21.0%. An ARC thickness of 80 nm gives an enhancement of 3.33%.

**Figure 6 nanomaterials-14-00324-f006:**
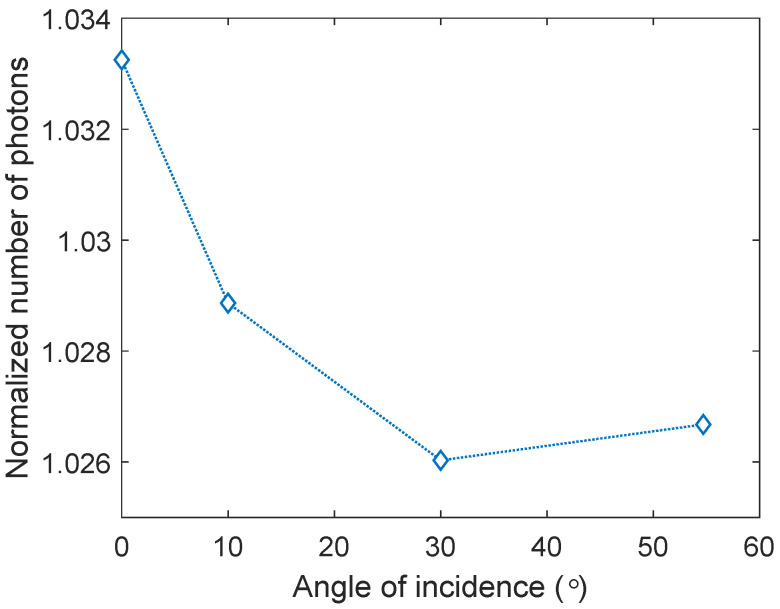
The normalized number of photons transmitted into c-Si for the optimized device parameters as a function of the angle of incidence of the light. Zero degrees corresponds to the flat substrate, while 54.7° corresponds to normal incidence on a textured c-Si solar cell.

**Figure 7 nanomaterials-14-00324-f007:**
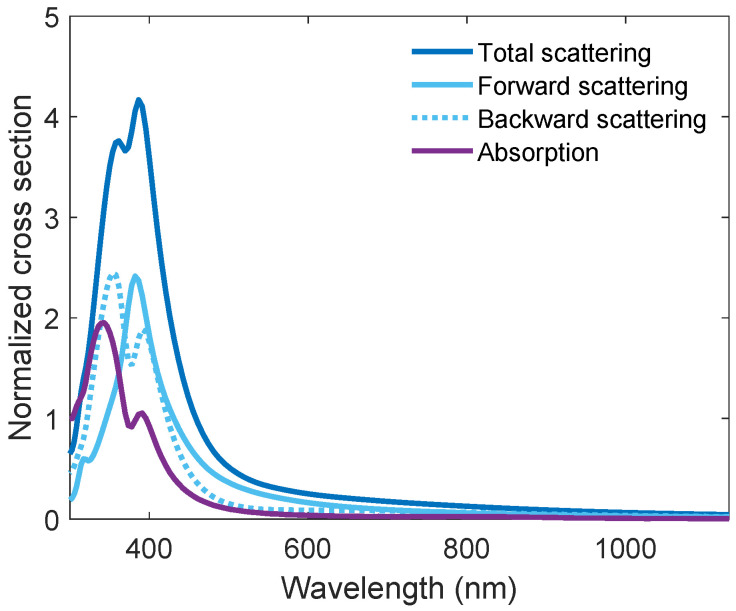
The scattering (dark blue) and absorption (purple) cross section plots of the Al nanoparticles. The light blue curves show the scattering cross section split into forward (solid) and backward (dashed) scattering.

**Figure 8 nanomaterials-14-00324-f008:**
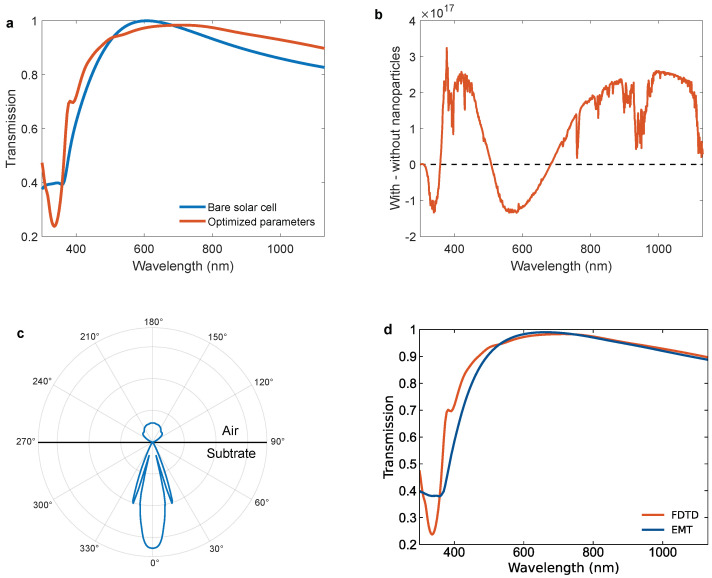
(**a**) The light source independent transmission into c-Si as a function of wavelength for a bare solar cell (blue) and for a solar cell with optimized parameters (orange). (**b**) The difference in the actual number of photons transmitted into c-Si with and without the optimized parameters as a function of wavelength. (**c**) The angular scattering distribution from the nanoparticle at a wavelength of 640 nm. (**d**) The transmission of light into c-Si as a function of wavelength for a solar cell with Al nanoparticles using the Fresnel equations with EMT (blue), and the FDTD simulation for the same system (orange).

**Table 1 nanomaterials-14-00324-t001:** The surface coverages simulated and the corresponding interparticle spacings for each of the nanoparticle sizes.

Surface Coverage	80 nm	100 nm	120 nm	140 nm
1.9%	520 nm	650 nm	780 nm	910 nm
2.6%	440 nm	550 nm	660 nm	770 nm
3.9%	360 nm	450 nm	540 nm	630 nm
6.4%	280 nm	350 nm	420 nm	490 nm
9.0%	236 nm	295 nm	354 nm	414 nm
12.6%	200 nm	250 nm	300 nm	350 nm
15.0%	183 nm	229 nm	275 nm	320 nm
18.0%	167 nm	209 nm	251 nm	292 nm
21.0%	155 nm	193 nm	232 nm	271 nm
34.9%	120 nm	150 nm	180 nm	210 nm

## Data Availability

The datasets generated and analyzed during the current study are available from the corresponding author upon reasonable request.

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
