# Peer review of "Enhancing Silicon Solar Cell Performance Using a Thin-Film-like Aluminum Nanoparticle Surface Layer"

_nanomaterials, 2024, doi:10.3390/nano14040324_

Round 1

Reviewer 1 Report

Comments and Suggestions for Authors

The authors showed an configuration for maximizing light in-coupling into a standard textured crystalline silicon solar cell by determining the optimal Al nanoparticle and anti-reflection coating parameters. This manuscript is somewhat interesting, but should be improved in several parts:

1.     The novelty of this manuscript is not clear enough. The authors should further highlight the novelty.

2.     More detailed effect and explanation of the working mechanism are suggested to be added.

3.     The details of simulations are missing. The authors should describe it as detailed as possible.

4.     How to further improve the performance by using the proposed strategy? The authors are suggested to give some comments.

5.     To make this manuscript more interesting and general, some papers should be cited (e.g., Journal of Materials Chemistry A 2019, 7, 1539-1547).

Comments on the Quality of English Language

Good.

Reviewer 2 Report

Comments and Suggestions for Authors

In this manuscript, Mirjam D. Fjell et al. used Al nanoparticles to maximize light in-coupling into a standard textured crystalline silicon solar cell for their tunable LSPR in the ultraviolet spectral region and low light absorption. The authors successfully achieved a significant increase in photon absorption efficiency, resulting in a notable enhancement of 3.3%. Furthermore, the prevailing mechanisms governing the interaction between light and matter were comprehensively elucidated. It would be better if the following details or specific points were added to this study.

1.     In perovskite solar cells, Al is known to easily diffuse when used as a device electrode, is this also the case here?

2.     What is the complete structure of c-Si solar cell in this strategy? Please provide.

3.     Here, Al nanoparticles are used. Will other nanoparticles such as Ag, Au, or oxide (Al2O3, SiO2, TiO2, SnO2) nanoparticles have similar effects?

4.     The inverted pyramid structure is widely used as a anti reflection structure in silicon batteries. Does the anti-reflection effect of the nanoparticles in this article match that of this structure?

5.     Can this Al nanoparticle anti-reflective layer be applied to other types of solar cells?
